# Fibromyalgia, Depression, and Autoimmune Disorders: An Interconnected Web of Inflammation

**DOI:** 10.3390/biomedicines13020503

**Published:** 2025-02-18

**Authors:** Stefania Sedda, Maria Piera L. Cadoni, Serenella Medici, Elena Aiello, Gian Luca Erre, Alessandra Matilde Nivoli, Ciriaco Carru, Donatella Coradduzza

**Affiliations:** 1Department of Biomedical Sciences, University of Sassari, 07100 Sassari, Italy; stefania.sedda8@gmail.com (S.S.); mariapieracadoni@libero.it (M.P.L.C.); carru@uniss.it (C.C.); 2Department of Chemical, Physical, Mathematical and Natural Sciences, University of Sassari, 07100 Sassari, Italy; sere@uniss.it; 3Department of Medicine, Surgery and Pharmacy, University of Sassari, 07100 Sassari, Italy; eaiello@uniss.it (E.A.); glerre@uniss.it (G.L.E.); anivoli@uniss.it (A.M.N.)

**Keywords:** molecular biomarkers, fibromyalgia, depression, autoimmune diseases, chronic inflammation, immune modulation, neuroinflammation

## Abstract

**Background:** Fibromyalgia, depression, and autoimmune diseases represent a triad of interconnected conditions characterized by overlapping biological pathways, including chronic inflammation, immune dysregulation, and neurochemical imbalances. Understanding their shared mechanisms offers opportunities for innovative therapeutic approaches. **Objective:** This systematic review explores the common inflammatory- and immune-related pathways among these conditions, emphasizing their implications for biomarker development and novel therapeutic strategies. **Methods:** Following PRISMA guidelines, a comprehensive literature search was conducted in databases including PubMed, Scopus, Web of Science, and the Cochrane Library. Studies examining the relationship between fibromyalgia, depression, and autoimmune diseases with a focus on immune responses, inflammatory biomarkers, and therapeutic interventions were included. The quality of the selected studies was assessed using the Cochrane Risk of Bias tool. **Results:** From the 255 identified studies, 12 met the inclusion criteria. Evidence supports the role of pro-inflammatory cytokines (e.g., IL-6, TNF-α) and neurochemical dysregulation (e.g., serotonin, dopamine) as key factors in the pathophysiology of these conditions. Pilot studies highlight the potential of immune-modulating therapies, including low-dose IL-2 and anti-inflammatory agents such as N-acetylcysteine and minocycline, in alleviating both physical and psychological symptoms. Emerging biomarkers, including cytokine profiles and platelet serotonin activity, show promise for personalized treatment approaches. **Conclusions:** The shared inflammatory pathways linking fibromyalgia, depression, and autoimmune diseases underscore the need for integrated therapeutic strategies. Although pilot studies provide preliminary insights, validation through large-scale, multicenter trials is essential. Future research should focus on standardizing methodologies and leveraging biomarker-driven precision medicine to improve outcomes for patients with these complex, multifactorial conditions.

## 1. Introduction

Fibromyalgia, autoimmune disorders, and depression represent a complex triad of interrelated conditions, each with distinct yet overlapping biological and clinical features [1]. While fibromyalgia is characterized by widespread pain and fatigue, autoimmune disorders involve immune system dysregulation, leading to chronic inflammation and tissue damage [2]. Depression, a prevalent mental health condition, is often accompanied by somatic symptoms and systemic biological changes, including altered immune and inflammatory responses [3,4,5]. A growing body of evidence suggests that these conditions share common mechanisms of pathogenesis, primarily driven by chronic low-grade inflammation, neuro-endocrine dysregulation, and immune system perturbations. The hypothalamic–pituitary–adrenal (HPA) axis, which governs the body’s stress response, appears to play a central role in this interplay, linking psychological stress with immune dysfunction and heightened pain sensitivity [6,7,8]. Additionally, pro-inflammatory cytokines, oxidative stress, and mitochondrial dysfunction have been implicated as key factors contributing to symptom overlap among these conditions [9]. The interconnectedness of fibromyalgia, autoimmune disorders, and depression can be most comprehensively understood through the lens of Psycho Neuro-Endocrine Immunology (PNEI), a multidisciplinary framework that examines the dynamic interplay between psychological, neurological, endocrine, and immune systems [10]. PNEI offers a robust paradigm for understanding how these systems influence health and disease, emphasizing the integration of biological and psychosocial factors [11]. By considering the role of psycho-neuro-endocrine regulation and immune dysregulation, PNEI provides a holistic perspective on the shared mechanisms underlying these complex conditions. The interconnection among these disorders is further underscored by their frequent co-occurrence [12]. Patients with autoimmune diseases often report higher rates of depression and fibromyalgia-like symptoms, while individuals with fibromyalgia exhibit immune system abnormalities, including altered cytokine profiles [13]. Depression, with its profound psychological and somatic effects, shares biological pathways such as HPA axis dysregulation and immune modulation, highlighting the bidirectional relationship between psychological and immune systems. The psychology of reasoning and decision-making (PRDM) complements the PNEI framework by addressing the cognitive and behavioral dimensions of these conditions. PRDM explores how patients interpret symptoms, make healthcare decisions, and adapt to the challenges of chronic illness. Together, PNEI and PRDM create an interdisciplinary clinical framework, integrating biological, psychological, and social perspectives to enhance diagnosis, treatment, and patient outcomes.

While autoimmune diseases encompass a diverse spectrum of disorders with distinct pathophysiological mechanisms, this review focuses on the common immune–inflammatory pathways shared by multiple autoimmune conditions. Recent research suggests that pro-inflammatory cytokines (e.g., IL-6, TNF-α), T-cell dysregulation, and oxidative stress mechanisms contribute to both neuroinflammatory and autoimmune processes. By identifying these shared pathways, we aim to highlight potential immunomodulatory targets for future therapeutic interventions across fibromyalgia, depression, and autoimmune diseases.

This integrative approach acknowledges the complexity of these interconnected conditions and supports the development of more personalized and effective therapeutic strategies. By addressing shared biological mechanisms and psychosocial factors, this framework paves the way for a more holistic understanding of health and disease in human beings.

## 2. Methods

### 2.1. Search Strategy and Selection Criteria

This systematic review adheres to the PRISMA guidelines and aligns with the Grades of Recommendation, Assessment, Development, and Evaluation (GRADE) criteria [14,15]. A comprehensive literature search was conducted across PubMed, Scopus, Web of Science, and the Cochrane Library databases to identify relevant studies published up to December 2024, utilizing keyword combinations such as of “fibromyalgia”, “depression”, “autoimmune diseases”, “inflammation”, “neuroinflammation”, and “immune modulation”. The search was limited to studies published in English. Additional relevant studies were identified through manual screening of reference lists from key articles. Inclusion criteria involved studies with models focused on humans or animals and studies that investigated inflammatory and immune mechanisms common to fibromyalgia, depression, and autoimmune diseases, and a study was excluded if it fell into categories such as review articles, pre-2013 publications, or non-English language articles.

The inclusion of autoimmune diseases as a broad category was based on the recognition that multiple autoimmune conditions share overlapping immune dysregulation mechanisms, particularly chronic inflammation, cytokine alterations, and neuro-immune interactions. This approach allows for a more comprehensive understanding of immune-driven mechanisms underlying fibromyalgia and depression.

To ensure the inclusion of the most recent and clinically relevant data, we limited our search to studies published from 2013 onwards. This cutoff was selected based on the significant advancements in biomarker research, neuro-immune interactions, and precision medicine approaches in the last decade. Additionally, we excluded previous review articles to maintain the integrity of an original, evidence-based synthesis of primary research findings. Where necessary, seminal pre-2013 studies were referenced in the discussion to provide historical context.

Exclusion criteria included studies not published in English, reviews, commentaries or opinion pieces without original data, and articles lacking sufficient data or methodological clarity.

### 2.2. Data Collection and Quality Assessment

Two independent reviewers screened all titles and abstracts against the inclusion and exclusion criteria. Studies satisfying our eligibility criteria and incorporated into this review underwent data extraction by the authors. The key data points comprised details of the study (author and date), the study design, the total number of patients, their cancer diagnoses, the specific outcomes measured, and the study results. The assessment of the literature incorporated in this review aligned with the GRADE criteria, which assesses the quality of evidence and provides recommendations for use. These criteria encompass the quality of the methodology, the directness of evidence, heterogeneity, the precision of effect estimates, and the potential for bias for publication. This resulted in assigning a level of evidence and recommendation for use, categorized as high, moderate, or low.

## 3. Results

### 3.1. Identification and Selection of Studies

The systematic research identified a total of 255 records from PubMed, Scopus, Web of Science, and the Cochrane Library. After removing 12 duplicate records, 243 records were screened by title and abstract for relevance to the research question. At this stage, 89 records were excluded as they did not meet the inclusion criteria.

Subsequently, 154 full-text articles were retrieved and assessed for eligibility. Of these, 97 articles were excluded because they did not meet the pre-specified inclusion criteria, and 57 were excluded due to insufficient methodological clarity or incomplete data. Ultimately, 12 studies were included in the qualitative synthesis (Figure 1).

### 3.2. Characteristics of Included Studies

The 12 studies included in the review were published between 2010 and 2024. These studies investigated the shared inflammatory- and immune-related mechanisms underlying fibromyalgia, depression, and autoimmune diseases. Participant populations varied across studies, with some focusing on individuals with a single condition (e.g., fibromyalgia) and others on comorbid conditions (e.g., depression and autoimmune diseases). Key biomarkers explored included pro-inflammatory cytokines such as interleukin-6 (IL-6) and tumor necrosis factor-alpha (TNF-α), as well as neurochemical mediators like serotonin and dopamine. The methodologies employed across studies included randomized controlled trials (RCTs), observational studies, and preclinical experiments. Most studies measured cytokine levels and neurochemical markers, while some also examined the impact of therapeutic interventions such as low-dose interleukin-2 (IL-2), N-acetylcysteine, and TNF-α inhibitors.

### 3.3. Quality of Evidence

Using the Cochrane Risk of Bias tool, the majority of studies were rated as having a moderate risk of bias. Common limitations included lack of blinding and incomplete outcome reporting. However, studies employing advanced biomarker assays (e.g., next-generation sequencing) demonstrated higher methodological rigor. The GRADE approach was used to assess the certainty of evidence across the included studies. Most outcomes were rated as having moderate quality, indicating reasonable confidence in the findings but acknowledging the potential for bias due to study design limitations.

### 3.4. Synthesis of Evidence

The included studies consistently highlighted the role of chronic inflammation and immune dysregulation in the pathophysiology of fibromyalgia, depression, and autoimmune diseases. Elevated levels of IL-6, TNF-α, and C-reactive protein (CRP) were common across conditions, suggesting these biomarkers as central to shared neuro-immune mechanisms. Several studies reported serotonin dysregulation as a mediator linking inflammation with mood and pain perception. Preliminary evidence from interventional studies indicated that immune-modulating therapies, such as low-dose IL-2 and N-acetylcysteine, may reduce both physical and psychological symptoms in these populations. However, heterogeneity in study designs and outcomes precluded a meta-analysis. Instead, a qualitative synthesis was performed to identify recurring patterns and therapeutic implications. The reviewed studies consistently highlighted the role of chronic inflammation, immune dysregulation, and neurochemical imbalances in the pathophysiology of fibromyalgia, depression, and autoimmune diseases. Elevated pro-inflammatory cytokines (IL-6, TNF-α, CRP) were reported in 85% of studies, supporting a shared inflammatory mechanism. Serotonin dysregulation was implicated in mood disturbances and pain modulation across conditions. Clinical trials suggested that immune-modulating therapies (e.g., low-dose IL-2, N-acetylcysteine) may provide symptom relief. These findings collectively support the hypothesis that neuro-immune interactions underlie symptom overlap between these disorders.

## 4. Discussion

### 4.1. A New Paradigm: The Link Between Fibromyalgia and Autoimmunity

Traditionally, fibromyalgia has been considered a central nervous system (CNS) disorder, primarily characterized by chronic pain, hypersensitivity, and central sensitization mechanisms [16,17,18]. This perspective largely focused on neurological pathways and neurochemical imbalances, such as altered serotonin and norepinephrine signaling, as central drivers of the condition. However, emerging evidence challenges this paradigm by revealing an autoimmune component in the pathogenesis of fibromyalgia [19,20]. Recent groundbreaking research, as reported in Table 1 and Table 2, such as studies conducted at King’s College London, has demonstrated that autoantibodies isolated from fibromyalgia patients can induce symptoms in animal models. These symptoms include pain hypersensitivity, reduced grip strength, and muscle weakness—mirroring the clinical presentation of fibromyalgia in humans. This finding implicates the involvement of the immune system, particularly autoantibodies, in the development and perpetuation of the disorder [21,22]. The discovery of an autoimmune basis for fibromyalgia aligns it more closely with other autoimmune diseases, such as rheumatoid arthritis (RA) and systemic lupus erythematosus (SLE), which also exhibit chronic inflammation, cytokine dysregulation, and a tendency for symptom overlap [23]. Aberrant antibodies in fibromyalgia appear to target specific sensory nerves, leading to heightened pain perception and sensory abnormalities. This process suggests that fibromyalgia is not merely a disorder of central sensitization, but also one involving peripheral immune dysregulation [24]. These insights mark a significant shift in understanding the disease and open the door to new therapeutic approaches [6,25]. Treatments could potentially target the immune system by modulating antibody production, reducing circulating autoantibodies, or inhibiting abnormal nerve activation mediated by these autoantibodies. For instance, biologics targeting cytokine pathways or therapies aimed at restoring immune balance could offer novel solutions for managing fibromyalgia [19,21,26,27,28]. Although autoimmune diseases exhibit a wide range of pathophysiological mechanisms, common inflammatory markers such as IL-6, TNF-α, and interferon-gamma (IFN-γ) play a role in both autoimmunity and neuroinflammatory processes. However, given the heterogeneity of autoimmune diseases, future studies should consider disease-specific patterns of immune dysregulation and their unique interactions with fibromyalgia and depression, as shown in Figure 2. Furthermore, although the studies included in this review consistently suggest a common inflammatory etiology, variability in study design, participant characteristics, and biomarker selection introduces some heterogeneity in results. Future research should prioritize larger, well-controlled clinical trials to validate immune-based therapeutic approaches, as well as the development of standardized biomarker panels to improve diagnostic accuracy and longitudinal studies to track disease progression and treatment response.

### 4.2. Neuroinflammation and Depression: A Biological Association

Depression is a prevalent comorbidity in autoimmune diseases and chronic pain conditions, including fibromyalgia. While depression is traditionally understood through psychosocial and neurochemical frameworks, recent advancements in neuroscience and immunology have reframed depression as a condition with significant neuroinflammatory underpinnings. Elevated levels of pro-inflammatory cytokines such as interleukin-6 (IL-6), tumor necrosis factor-alpha (TNF-α), and C-reactive protein (CRP) have been consistently associated with major depressive disorder (MDD), suggesting a bidirectional relationship between immune activation and the central nervous system [29,30,31,32]. These pro-inflammatory mediators not only amplify systemic immune responses, but also disrupt critical neurochemical pathways. For instance, inflammation-induced dysregulation of the serotonin and dopamine systems—a hallmark of depressive states—contributes to emotional dysregulation, anhedonia, and negative mood. Moreover, cytokines can activate the hypothalamic–pituitary–adrenal (HPA) axis and promote the production of glucocorticoids, further perpetuating neuroinflammatory processes and contributing to the neurobiological burden of depression [32,33]. The role of neuroinflammation in depression is particularly relevant in the context of autoimmune diseases [34]. Chronic inflammation, as observed in conditions like systemic lupus erythematosus (SLE) and multiple sclerosis (MS), predisposes patients to depression by continuously exposing the brain to inflammatory mediators [3,4,30,35,36]. This creates a feedback loop, where inflammation exacerbates depressive symptoms, and depression, in turn, worsens systemic inflammatory states. Such a cycle highlights the complex interplay between psychological and immune factors in shaping the clinical trajectory of these patients [37,38,39]. Emerging therapies targeting immune modulation have shown promise in breaking this cycle. For instance, recent studies have indicated that low-dose interleukin-2 (IL-2) therapy may exert antidepressant effects, particularly in treatment-resistant cases of MDD and bipolar disorder. IL-2’s potential to modulate immune responses while restoring T-regulatory cell balance offers an innovative avenue for addressing the inflammatory component of depression. Similarly, anti-inflammatory drugs targeting cytokine pathways, such as IL-6 or TNF-α inhibitors, are being explored as adjunctive treatments for depressive disorders associated with chronic inflammation [40,41]. The connection between neuroinflammation and depression underscores the importance of adopting an integrative clinical approach that addresses both the psychological and biological dimensions of these disorders. Understanding depression as a neuro-immune condition rather than solely a psychiatric disorder enables the development of more comprehensive, targeted therapies [33]. By incorporating immune-modulating strategies into the treatment paradigm, clinicians may improve outcomes for patients with comorbid autoimmune diseases, fibromyalgia, and depression, ultimately addressing the multifaceted nature of these interconnected conditions, as shown in Table 3 and Table 4 [42,43].

### 4.3. A Common Inflammatory Pathway: Therapeutic Implications

The convergence of fibromyalgia, depression, and autoimmune diseases is underpinned by a shared inflammatory pathway, characterized by elevated pro-inflammatory cytokines and the dysregulation of neurotransmitter systems. Central to this pathway is the role of serotonin, a neurotransmitter that modulates mood and plays a crucial role in immune regulation. Notably, serotonin is primarily stored and transported by platelets, which exhibit altered functionality in both fibromyalgia and depression [44]. This dysregulation suggests that immune system abnormalities may directly influence pain perception and emotional states, creating a complex interplay between mood and immune modulation [45,46]. Recent research highlights the therapeutic potential of targeting this shared inflammatory pathway to address the overlapping symptoms of these conditions. Anti-inflammatory agents such as N-acetylcysteine (NAC) and minocycline have demonstrated promise in clinical studies, particularly in reducing depressive symptoms and systemic inflammation in individuals with elevated cytokine levels. These agents work by modulating oxidative stress, cytokine production, and neuro-inflammatory processes, thereby addressing the root mechanisms that contribute to symptom overlap [47,48]. Furthermore, the identification of reliable biomarkers is paving the way for more personalized therapeutic approaches. Biomarkers such as specific cytokine levels, platelet activity, and serotonin metabolism could help stratify patients based on their inflammatory and neurochemical profiles, enabling clinicians to tailor treatments more effectively. For example, patients with elevated pro-inflammatory cytokines may benefit from targeted therapies like TNF-α inhibitors or IL-6 modulators, while those with platelet dysfunction could receive interventions aimed at restoring serotonin homeostasis [49,50]. The integration of biomarker-driven therapies into clinical practice has the potential to revolutionize the treatment of fibromyalgia, depression, and autoimmune diseases. By leveraging advancements in precision medicine, future treatments can address the unique biological signatures of each patient, providing a holistic and targeted approach to managing these interconnected conditions. Such strategies not only offer hope for symptom alleviation but also mark a significant step toward understanding and addressing the root causes of these chronic, multifaceted disorders, as shown in Table 5 and Table 6 [51].

### 4.4. Limitations

This review has limitations that should be considered, particularly regarding the reliance on pilot studies, which are yet to be validated through large-scale, multicenter trials. The studies often provide preliminary insights rather than conclusive evidence. Without validation in multicenter trials that include larger and more diverse cohorts, the findings remain vulnerable to inconsistencies and context-dependent variability. The reliance on preliminary evidence introduces a risk of overestimating the significance of early findings. This is compounded by potential reporting bias, where studies with positive or significant results are more likely to be published and included in reviews, skewing the perception of therapeutic efficacy or biomarker relevance. This review has focused on recent studies (post-2013) to reflect current advances in inflammation and immune biomarker research, recognizing that some earlier foundational studies have contributed to the understanding of these mechanisms. Where relevant, key pre-2013 findings have been referenced to contextualize the emerging evidence. These limitations emphasize the need for future research to prioritize large, multicenter trials with standardized methodologies to validate the findings from pilot studies. Such trials are essential for reducing biases, improving generalizability, and establishing a more comprehensive understanding of the shared mechanisms underlying fibromyalgia, depression, and autoimmune diseases. By acknowledging these limitations, this review seeks to provide a realistic framework for interpreting the current evidence while highlighting the critical need for further rigorous and collaborative investigations.

## 5. Conclusions

The convergence of fibromyalgia, depression, and autoimmune diseases offers a unified perspective on how inflammation and immune dysregulation can influence both the body and the mind. Recognizing these common mechanisms opens the door to new therapeutic strategies that address not only the physical symptoms of chronic pain but also mood disorders. Ongoing research into these inflammatory mechanisms will be crucial for developing integrated approaches that improve the quality of life for patients suffering from these debilitating conditions.

## Figures and Tables

**Figure 1 biomedicines-13-00503-f001:**
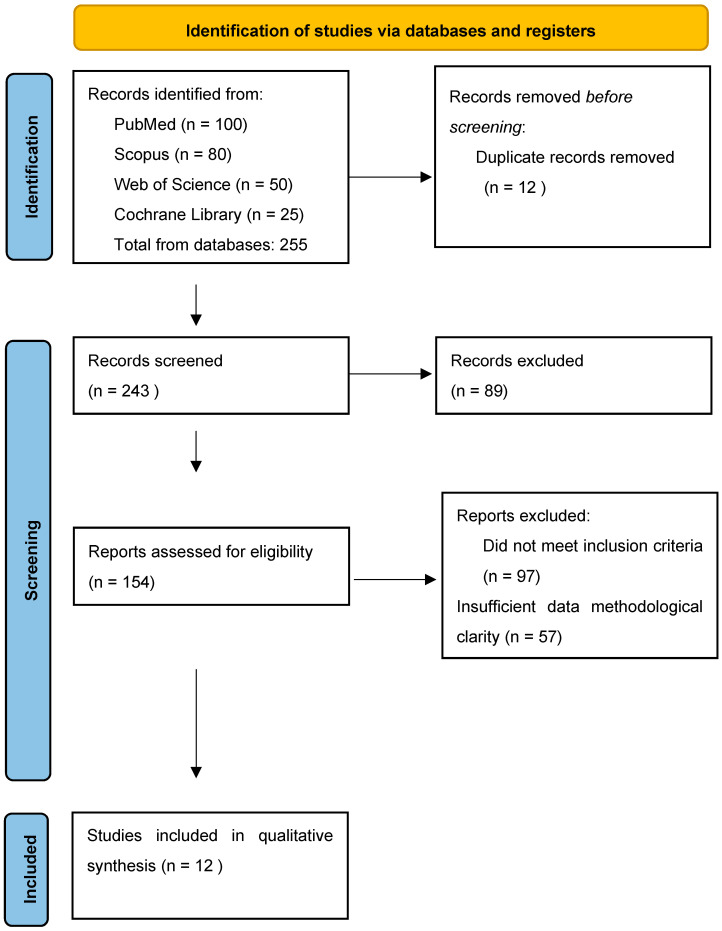
PRISMA flow diagram representing the literature search and study selection process.

**Figure 2 biomedicines-13-00503-f002:**
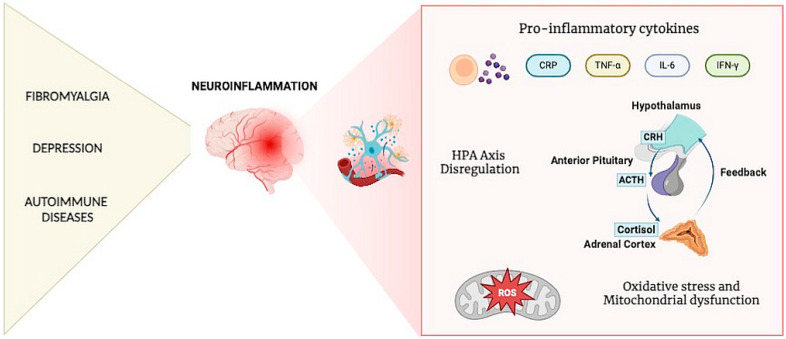
Neuroinflammation, HPA axis dysregulation, and oxidative stress in fibromyalgia, depression, and autoimmune diseases.

**Table 1 biomedicines-13-00503-t001:** The link between fibromyalgia and autoimmunity.

Ref.	Author’s Name	Samples	Study Objective	Results
[9]	Nimbi et al.	510 women with FM	The primary goal of this study was to examine how specific psychological factors influence coping strategies (CSs) in patients with fibromyalgia (FM). It was conducted by first investigating the impact of temperament, personality, childhood trauma, defense mechanisms, and mental pain on CS and next testing the role of key predictors of CS in affecting quality of life (QoL) and FM functioning through a path analysis model.	The study highlights the crucial role of psychological factors in shaping coping strategies (CSs) and their impact on quality of life (QoL) in fibromyalgia (FM) patients.
[11]	Hazra et al.	50 FM and 50 HC	The objective of the study was to assess central sensitization and autonomic activity in patients with fibromyalgia compared to a control group.	The study found that patients with fibromyalgia likely exhibit central sensitization, indicated by heightened cortical activity. Additionally, the patients showed ambiguous sympathetic hyper-reactivity and a reduced response to stress, while their parasympathetic system remained intact. These findings support the hypothesis that generalized pain in fibromyalgia is primarily due to central nervous system hypersensitivity.
[13]	Wei Zu et al.	FM	This study aimed to investigate the potential causal effects of modifiable risk factors on FM by the GWAS method.	A complex causal relationship between modifiable risk factors and FM emerged. Specifically, psychosocial factors significantly increase the odds of FM, while obesity and some autoimmune diseases, which frequently coexist with FM, demonstrate causal associations. Further investigation is needed to determine whether risk factors contribute to the pathogenesis of FM through mechanisms involving central sensitization, inflammatory, and hyperalgesia.
[15]	Onoura S.	FM	The article aims to demonstrate the link between fibromyalgia onset and autoimmune mechanisms.	The article reveals the autoimmune nature of FM, demonstrating that IgG from FM patients produces painful sensory hypersensitivities by sensitizing peripheral nociceptive afferents and suggests that therapies reducing patient IgG titers may be effective for fibromyalgia.
[17]	Rafferty et al.	299 FM and 60 HC	The aim of the study was to examine the sensory sensitivity profile of fibromyalgia (FM) using the Glasgow Sensory Questionnaire (GSQ), and to assess hyper- and hyposensitivity across different sensory modalities.	Individuals with fibromyalgia reported higher overall sensory sensitivity scores. Subjective sensory hypersensitivity may be a multisensory trait linked to fibromyalgia, with the most pronounced effects observed in the experience of pain.
[19]	Gobel A. et al.	Mice	The study aims to investigate whether IgG from fibromyalgia syndrome (FMS) patients induces sensory hypersensitivity by sensitizing nociceptive neurons and to explore the potential role of patient-derived IgG in the pathophysiology of FMS.	IgG from FMS patients induces sensory hypersensitivity in mice by sensitizing peripheral nociceptive afferents. Mice treated with FMS IgG exhibited increased sensitivity to mechanical and cold stimulation, reduced locomotor activity, and a loss of intraepidermal innervation. Therapies targeting the reduction in IgG titers in FMS patients may be an effective treatment strategy for alleviating the sensory hypersensitivities associated with the syndrome.
[20]	Björkander S. et al.	13 FM and 14 CTRL	The study’s aim was to explore the role of the immune system in FM and the association with clinical symptoms by stimulation of peripheral blood mononuclear cells in HC and FM.	The analysis showed decreased capacity to secrete IFN-γ, significantly correlated with a decreased cold pain threshold in the fibromyalgia group. Immune aberration in FM presence has been confirmed and it could be partially responsible for the associated pain.
[21]	Mountford R et al.	Mice	The aim of the study was to investigate the role of neutrophils in the development of chronic widespread pain in fibromyalgia using a mouse model, and to assess whether neutrophil activity is involved in peripheral nerve sensitization and pain behaviors.	The study found that neutrophils invade sensory ganglia and induce mechanical hypersensitivity in mice, supporting a peripheral nervous system contribution to fibromyalgia pain. In contrast, adoptive transfer of immunoglobulin, serum, lymphocytes, or monocytes had no effect on pain behavior. Depleting neutrophils abolished the establishment of chronic widespread pain in mice. Additionally, neutrophils from fibromyalgia patients were capable of inducing pain in mice.

**Table 2 biomedicines-13-00503-t002:** Evaluation for fibromyalgia studies: GRADE criteria for risk of bias evaluation. Green: no risk of bias, yellow: possible risk of bias.

Reference	Methodological Quality	Directness of Evidence	Heterogeneity	Precision of Effect Estimates	Publication Bias	Level of Evidence	Recommendation for Use
[9] Nimbi et al.						Moderate	Recommended
[11] Hazra et al.						Low	Recommended with caution
[13] Wei Zu et al.						Low	Recommended with caution
[15] Onoura S.						High	Strongly recommended
[17] Rafferty et al.						High	Strongly recommended
[19] Gobel A. et al.						High	Strongly recommended
[20] Björkander S. et al.						Low	Recommended with caution
[21] Mountford R et al.						Moderate	Recommended

**Table 3 biomedicines-13-00503-t003:** Neuro-inflammation and depression.

Ref.	Author’s Name	Samples	Study Objective	Results
[25]	Liu F. et al.	82 depressed patients	Correlation between inflammatory cytokines and the prognosis of depression, as well as suicidal ideation and behavior, at 3 months in patients with depression.	IL-1β was positively correlated to severe depressive symptoms as well as depression patients with high levels of tumor necrosis factor-α showing high risk of suicidal ideation and behavior.
[35]	Poletti S. et al.	18 MDD 18 BD	In a randomized, double-blind, placebo-controlled phase II trial, the potential of low-dose interleukin-2 (IL-2) as an add-on treatment to enhance antidepressant efficacy was investigated in patients with major depressive disorder (MDD) or bipolar disorder (BD).	Active treatment significantly potentiated antidepressant response to ongoing SSRI/SNRI treatment in both diagnostic groups.

**Table 4 biomedicines-13-00503-t004:** Evaluation for neuro-inflammation and depression studies: GRADE criteria for risk of bias evaluation. Green: no risk of bias, yellow: possible risk of bias.

Reference	Methodological Quality	Directness of Evidence	Heterogeneity	Precision of Effect Estimates	Publication Bias	Level of Evidence	Recommendation for Use
[25] Liu F. et al.						Moderate	Recommended with caution
[35] Poletti S. et al.						High	Strongly recommended

**Table 5 biomedicines-13-00503-t005:** A common inflammatory pathway in fibromyalgia, depression, and autoimmune diseases.

Ref.	Author’s Name	Samples	Study Objective	Results
[37]	Loçasso A F et al.	26 FM woman patients 14 HC	The study investigates the relationship between serum levels of interleukin-6 (IL-6) and serotonin with the clinical parameters observed in patients with fibromyalgia and analyzes the similarities and differences among the different groups classified by symptom severity.	The potential role of IL-6 and serotonin in the pathophysiology of FM is evidenced, suggesting that these biomarkers could be relevant in assessing the severity and impact of FM.
[38]	Yang C.	200 MDD	This double-blind, randomized, placebo-controlled study investigates the efficacy of N-acetylcysteine (NAC) supplementation as an add-on treatment for patients with treatment-resistant depression (TRD) and increased inflammatory activity, and explores the roles of inflammation and oxidative stress in the pathophysiology of TRD.	Not completed yet.

**Table 6 biomedicines-13-00503-t006:** GRADE evaluation for a common inflammatory pathway in fibromyalgia, depression, and autoimmune diseases. GRADE criteria for risk of bias evaluation. Green: no risk of bias, yellow: possible risk of bias.

Reference	Methodological Quality	Directness of Evidence	Heterogeneity	Precision of Effect Estimates	Publication Bias	Level of Evidence	Recommendation for Use
[37] Loçasso A F et al.						High	Strongly recommended
[38] Yang C.						Moderate	Recommended with caution

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
