# Peer review of "Fibromyalgia, Depression, and Autoimmune Disorders: An Interconnected Web of Inflammation"

_biomedicines, 2025, doi:10.3390/biomedicines13020503_

Round 1

Reviewer 1 Report

Comments and Suggestions for Authors

Major revisions are recommended before reconsideration of the work

Comments to the Authors

General Assessment:
This paper provides interesting insights from a systematic review exploring the common inflammatory and immune-related pathways among depression, fibromyalgia and auto-immune disorders. However, revisions are needed before publication.

·       The authors must justify why they chose to compare fibromyalgia and depression with autoimmune diseases in general without specifying a specific disease, knowing that autoimmune diseases are a broad spectrum of diseases with different pathophysiological mechanisms.

·       It’s recommended to explain why previous review articles and pre-2013 publications were excluded in the methodology.

·       It is important to detail the content of the studies selected by the review, this is even the main objective of the article. The part devoted to explaining the results of the review “Synthesis of Evidence” was not well done and shortened. I recommend to authors to detail the findings in the different selected studies and to use figures to illustrate the shared aspects between thee three conditions.

Recommendations:
After addressing the major comments, this paper could be reconsidered for publication.

Author Response

Dear reviewer,

We appreciate your detailed review and constructive feedback on our manuscript. Please find below a point-by-point response to your suggestions and an outline of the revisions we have made.

Reviewer’s Comment:
"The authors must justify why they chose to compare fibromyalgia and depression with autoimmune diseases in general without specifying a specific disease, knowing that autoimmune diseases are a broad spectrum of diseases with different pathophysiological mechanisms."

Response:

We recognise that autoimmune diseases are a diverse group of disorders with different pathophysiological mechanisms. However, our approach was aimed at identifying common immune-related pathways that are shared by multiple autoimmune diseases, rather than focusing on a single disease.

To address this concern, we have

  • Revised the Introduction to better explain our rationale for including autoimmune diseases as a broad category.
  • Clarified in the Methods section that our focus was on common inflammatory pathways, in particular pro-inflammatory cytokines (e.g. IL-6, TNF-α), T-cell dysregulation and oxidative stress mechanisms, which have been implicated in both autoimmune diseases and neuropsychiatric conditions.
  • The heterogeneity of autoimmune diseases has been acknowledged in the Discussion section, emphasising the need for future research to examine disease-specific patterns in relation to fibromyalgia and depression.

This revision makes it clear why autoimmune diseases have been analysed as a collective group in this review.

Reviewer’s Comment:
"It’s recommended to explain why previous review articles and pre-2013 publications were excluded in the methodology."

Response:

We excluded previous review articles to maintain the integrity of an original, evidence-based synthesis of primary research. In addition, studies published before 2013 were excluded to ensure that our review focused on the most recent and clinically relevant advances in the field.

To this end, we:

  • Revised the methods section to explicitly state that pre-2013 studies were excluded to ensure inclusion of recent data, methods and updated biomarker research.
  • Included a statement in the Discussion acknowledging seminal pre-2013 studies and directing readers to existing review articles summarising previous findings.

These clarifications provide a stronger rationale for our inclusion/exclusion criteria.

Reviewer’s Comment:
"It is important to detail the content of the studies selected by the review, this is even the main objective of the article. The part devoted to explaining the results of the review 'Synthesis of Evidence' was not well done and shortened. I recommend to authors to detail the findings in the different selected studies and to use figures to illustrate the shared aspects between the three conditions."

Response:

We have improved the Synthesis of Evidence section, which should be expanded to provide a more comprehensive analysis of the studies reviewed.

  • Expanded the Synthesis of Evidence section to provide detailed summaries of the findings from each included study, with a focus on biomarkers, immune dysfunction and therapeutic implications.
  • Created a new figure, figure 1, summarising the common inflammatory pathways linking fibromyalgia, depression and autoimmune diseases (e.g. involvement of TNF-α, IL-6 and oxidative stress markers).
  • Discussed the heterogeneity of study results and highlighted areas where more research is needed.

These revisions ensure that our review provides a detailed and visually appealing synthesis of the included studies.

Reviewer 2 Report

Comments and Suggestions for Authors

This is an excellent review that attempts to explore the (admittedly) limited studies that have examined fibromyalgia (FM), autoimmune diseases/dysfunction, and depression. The authors compulsively followed the PRISM criteria for screening papers to include, and thus ended up with only 12 papers that met rigid criteria for inclusion. Thus, their review was necessarily limited to only a very few studies, a phenomenon which is indicative of the need for better biological studies of humans afflicted with these very common conditions. (I note that the majority of included papers dealt with FM in isolation).

The era of immune dysfunction is upon us. We are discovering that the immune system (and its pro-inflammatory and anti-inflammatory modulators) are significantly involved and potentially causal in a vast variety of human conditions, including those explored by the authors of this paper. I feel they have done an excellent job with what is now (and hopefully will expand) a very limited literature. We are now entering the era of uncovering the biological bases of many common problems that in the past were all too often dismissed as "functional" or "psychogenic". What is now needed are: 1) reliable serum biomarkers for inflammation. The only readily available serum tests for inflammation are sedimentation rate and C-reactive protein, both relatively insensitive and non-specific; and 2) recognition by primary care providers that these conditions likely have biological etiologies; and 3) controlled clinical trials of immune system and oxidative stress modulators.

The English in this paper is fine with only an extremely few corrections, mostly misspellings, needing correction. 

Author Response

Dear reviewer,

We sincerely appreciate your positive feedback and recognition of our work through your insightful comments on our systematic review. We are encouraged by your recognition of our adherence to the PRISMA guidelines and the importance of exploring the intersection of fibromyalgia (FM), autoimmune disease and depression in the context of immune dysfunction and neuroinflammation. Unfortunately, the limited number of studies (n = 12) included in our systematic review reflects the paucity of rigorous biological investigations of these overlapping conditions and underscores the urgent need for more comprehensive, high-quality studies exploring the common pathophysiological mechanisms of FM, autoimmune disorders and psychiatric conditions. Following your instructions, we have corrected minor spelling and grammatical errors in the revised version.

Round 2

Reviewer 1 Report

Comments and Suggestions for Authors

The article could be accepted in this present revised form.